# The Development and Psychometric Properties of an Education Well-Being Scale among Elementary School Students

**DOI:** 10.3390/healthcare9081045

**Published:** 2021-08-13

**Authors:** Po-Lin Chen

**Affiliations:** Department of Counseling Psychology, Chinese Culture University, Taipei 111, Taiwan; cbl4@uluve.pccu.edu.tw

**Keywords:** education well-being, education well-being scale, psychometric, elementary students

## Abstract

The purpose of this study is to develop a tool that can measure the educational well-being of elementary school students and to analyze the fit of the developed instrument. The measurement tool was based on four levels which include: (1) parents’ support, (2) teachers’ care, (3) students’ care in school, and (4) peer cooperation. It also had eight facets which include: (1) basic need, (2) family companionship, (3) happy learning, (4) teacher-student interaction, (5) safety protection, (6) school rules, (7) peer learning, and (8) peer interaction. The participants of this study were grades 5 and 6 students in Taiwan. Pre-testing with 197 grade 6 students was done to assess the validity and reliability of the developed scale. For the formal study, a total of 960 grade 6 students and 834 grades 5 and 6 students were recruited to join. The data collected underwent item analysis, reliability estimation, and confirmatory factor analysis. The results of the analyses were as follows: (1) the confirmatory factor analysis supported the four latent factors of the educational well-being scale; (2) the Cronbach’s alpha ranged from 0.73 to 0.89 for the elementary school students; (3) the cross-validation analysis with split-half samples implies that the study had a well-constructed stability model and that the scale had construct validity; and (4) the average scores of grade 5 students on the 8 facets and their overall score on educational well-being were all significantly higher than that of grade 6 students. Finally, several suggestions were proposed for future studies based on the results obtained.

## 1. Introduction

Educational well-being in UNESCO’s Education for All (EFA) movement and the UN’s Decade of Education for Sustainable Development advocates that governments should believe that education plays an important role in the pursuit of well-being. Also, during the UN Convention on the Rights of the Child (UNCRC) [1,2], it was proposed that all educational forms, especially school education, should play a key role in the cultivation of well-being for adults and children and the ability of children to obtain a good life. The UN International Children’s Emergency Fund (UNICEF) [3] has put forward six facets of children’s well-being which are: (1) material welfare, (2) health and children, (3) education well-being, (4) family and peer relation, (5) actions and dangers and (6) subjective sense of well-being. They also provided the 36 quantifiable and comparable children indicators as policies and objectives for the improvement of children’s well-being.

It is of great importance to find ways on how to maximize children’s well-being in education to help them acquire the needed skills and ability to achieve well-being, comfort, and security in life [4]. Education exerts a great impact on well-being which is of three levels: (1) direct impact which assists people in developing the ability and resources that can influence their well-being; (2) indirect impact which produces results that can help people thrive and strengthen their resilience in a crisis; and (3) cumulative impact which influences people’s social and economic environment in their lifetime [5].

Educational well-being is described as the psychological disposition of learners, educators, and parents about educational policies, implementation of policies, and implementation plan and effect. When learners, educators, and parents are satisfied with the development, reform, and implementation of education, it is considered a kind of education well-being [6].

This study mainly follows the orientation of positive psychological research based on the education well-being proposed by previous literature [7,8,9,10] which chose elementary students as participants in developing an education well-being scale. This scale features good psychometric that can be used as a reference for students’ self-examination and the evaluation of school governance. Also, it can be utilized to assess the propriety of evidence-based decision-making, as a reference for policy-making, to determine the effectiveness of resource investment, and as a source of data for schools during their follow-up program planning and interlinked strategy formulation [11]. In the succeeding paragraphs, the development, connotation, and measure of educational well-being are analyzed and discussed, which were then used to design the current paper’s measurement scale for educational well-being.

The phrase “educational well-being” currently has no specific definition. The reason may be that each person has different indicators and degrees of well-being in education [4] According to Wu [9], educational well-being refers to a situation in which a person feels satisfied or happy with his or her education. It consists of two elements namely the core elements and the stimulative elements. The core elements, consisting of health, well-being, and hope, are the essence of educational well-being which belong to the most basic aspects of satisfaction. The stimulative elements are the impetus and facilitator, which help individuals develop a sense of well-being. Based on this, educational well-being can be evaluated subjectively and objectively. Subjective evaluation includes the perception of individuals and their level of satisfaction on educational well-being; while objective evaluation includes objective data on the individual’s rate of illiteracy, level of the quality of education, and educational attainment. In this study, the subjective evaluation of the educational well-being of students was used.

Wu [9] proposed that educational well-being includes four core concepts: (1) educational satisfaction, (2) educational achievement, (3) learning ability, (4) self-transcendence, and (5) positive thinking. Educational satisfaction is founded on the premise that educational action or teaching content and method should be based on the understanding of the individual differences and needs of learners. Only with learning satisfaction can one feel educational satisfaction and well-being. Educational achievement refers to the knowledge, ability, and attitude acquired by learners in the process of being taught and the development of their potential. Learning ability entails that each learner is different and possess a variety of learning ability that can not only transform themselves but can also change the society. Self-transcendence means that learners have the capacity to achieve self-realization, have the courage to explore, and have the ability to think and judge which allows them to experience the fun of learning in education and increase their well-being. Positive thinking indicates that when learners encounter difficulties, they face them with a positive attitude, seek solutions, take positive actions, and exert positive energy. Antonova et al. [7] conducted a survey on the educational well-being of students in Italy and Russia to assess students’ perception of the school environment (how the school helps in improving skills or abilities), their satisfaction with the school environment (such as the interaction with teachers), their freedom from corporal punishment, and the safety protection they receive in school. Hooker [8] point out that, the role of the teacher that has significant influence on promoting and developing student well-being in an educational setting that is crucial for them to be productive citizens. Lin [4] summarized the “well-being index” of Japan, including its subjective measurement which consisted of the following aspects: family, community, health management, curriculum and teaching, teachers and students, peers, school, and time allocation.

Adopted from previous relevant studies [7,8,9,10], this research explored educational well-being in four levels including parents, teachers, peers, and schools. Using these four levels, the indicators for the measurement tool were constructed downwards. Parents pay attention to children’s basic needs and family companionship. Teachers are concerned about the children’s learning experience and teacher-student interaction. Peers involve the children’s interactions with others and the assistance they receive. The school includes the safety protection they provide to the students and the rules and regulations they implement.

Besides, the Glück macht Schule in Austria aim at self-expression and social responsibility fulfillment and focus on happiness of life, psychological health, ability to love yourself, nutrition and body health, and exercise and body [12]. Wellington College in the UK also offers courses for achieving psychological health, positive relationship, vision, engagement, sustainable development and meaningful life [13]. After evaluation previous literature, it is realized that there is a lack of dimensions and tools for assessing educational well-being. As a result, the following indicators are being proposed.

### 1.1. Parents’ Support

In the process of interaction between individuals and their families, they can get psychological or physical assistance to ease the physical and psychological impact caused by pressure and improve their adaptation to life. Family relationships affect students’ mental health and feelings of happiness. Students with a better family relationship, parents’ support, and complete family structure have better psychological happiness and physical health. Parents’ support can be summarized into two:

#### 1.1.1. Basic Need

Parents’ support meets the physical, emotional, and life needs of students.

#### 1.1.2. Family Companionship

Parents’ support nurtures the relationship of students with their families and allows them to get involved in their schoolwork.

### 1.2. Teachers’ Care

During the interaction between teachers and students, teachers use words or examples to express their care and love for students, through this, students could feel their intentions. This makes learning a happier experience.

#### 1.2.1. Happy Learning

Good teacher-student relationship could result in improved students’ learning attitude and well-being. When teachers take the initiative to share their feelings with students and treat each of them equally, they develop a close and harmonious relationship. This makes students more likely to have a better learning attitude and feeling of well-being [14,15]. A school that fosters a happy atmosphere could make children excited to go to school. They would also expect that the school is a fun and happy place because the method of teaching makes learning a happy experience. A happy learning experience could be established when teachers interact with the students in class and give guidance depending on their needs, which helps students experience joy in the pursuit of knowledge.

#### 1.2.2. Teacher-Student Interaction

A good teacher-student interaction is one wherein the students feel that they can be friends with their teachers because the teachers genuinely care about their overall well-being regardless of place and time. This interaction allows students to get close to the teachers, which makes them enjoy the class.

### 1.3. Students’ Care in School

Aside from the equipment and facilities available for students in school, the school must provide a safe and comfortable learning environment. Adherence to good standards should be done to ensure the happy and healthy growth of students.

#### 1.3.1. Safety Protection

Gronna and Chin-Chance [16] pointed out that in a school environment with low safety levels, teachers need to spend more time in maintaining classroom discipline, which disturbs the learning and teaching activities and affects the students’ learning performance. Also, the students will be less able to concentrate on learning because of fear of being victims. An unsafe school will not be able to facilitate learning. The school should always consider each student’s overall personality and must provide a safe and barrier-free learning environment. This is an important task of the campus safety management.

#### 1.3.2. School Rules 

The school should set clear, specific, and reasonable rules to cultivate students’ self-discipline and self-respect. This will help students to clearly grasp the pattern of good behavior and know what to follow.

### 1.4. Peer Cooperation

When students can get along with peers, they obtain physical and mental comfort. Peers also help solve the difficulties that students face and improve their learning capacity and efficiency. Gifford-Smith and Brownell [17] pointed out that in the process of children’s growth, “peer interaction” and “group experience” exert great influence, and “peer” and “environment” are regarded as the key factors that affect children’s development. Good interpersonal relationships will have a positive impact on children’s emotional adaptation and work performance later in life [18].

#### 1.4.1. Peer Learning 

A harmonious class atmosphere and good interaction among students and teachers could develop good interpersonal communication skills, improve competitiveness, and facilitate happy learning. Hanafin and Brooks [19] found that children viewed their peers as an important factor, which is second only to family in their well-being.

#### 1.4.2. Peer Interaction 

Good influence among peers could help a student do better at school. They could remind each other of their homework and use their spare time to finish school projects together. By helping each other manage their time properly, tension, anxiety, and pressure from school could be reduced and conflicts between teachers and parents could be avoided. Through this, the educational well-being of children could be enhanced.

This study developed a measurement tool to determine the educational well-being of students. A pre-test was first conducted, followed by item and credibility analyses. Next, formal samples were collected and confirmatory factor analysis was used to test whether the structural model of factors was supported by the empirical data obtained. The test of measurement invariance was also used to evaluate whether the developed measurement tool has factor loading identity, intercept identity, and residual identity.

This study also explored whether there is a difference in the educational well-being of students belonging to different grade levels. The study by Argly and Lu [20] pointed out that well-being decreased as age and grade level increased. As one gets older, individual experience increases and accumulates which changes one’s perception of well-being [21,22]. In this study, students in elementary grades 5 and 6 were recruited as participants. Although they are considered senior elementary students, some of them are about to enter the adolescence precursor preparation stage while some have already entered the stage of puberty. Six graders will also enter middle school after school which will transition them from the elementary stage to the adolescent stage. Students’ experiences with physiological changes, school pressure, and the external environment vary from each other; thus, their perceptions of educational well-being are also different. Therefore, this study investigated whether there is a significant difference between the educational well-being of students in elementary grades 5 and 6.

Based on the above, this study aims to answer the following questions:

(1)Is the measurement tool for educational well-being a potential variable structure of second-order single factor?(2)What are the psychometric characteristics of the measurement tool for educational well-being?(3)Are there any differences in the educational well-being among students of different grade levels?

## 2. Materials and Methods

### 2.1. Participants

A total of 204 grade 6 students from an elementary school in Taiwan were selected to participate in the pre-testing through convenient sampling. Seven samples with more than 2 invalid answers were excluded. The total valid samples collected for pre-testing was 197 with a recovery rate of 96.6%.

For the formal test, grade 6 students from several schools in Taiwan were chosen as participants through stratified cluster sampling and purposive sampling. The schools were clustered based on size: large (with more than 60 classes), medium (25 to 59 classes), and small (24 classes). A total of 960 valid questionnaires were included, excluding those with incomplete and incorrect answers. The scale items, reliability, and validity of the valid questionnaires were then analyzed.

Another batch of participants from elementary grades 5 and 6 was chosen through convenience sampling. Similarly, questionnaires with incomplete and incorrect answers were excluded. A total of 530 valid questionnaires were obtained: 423 (63.5%) from grade 5 and 304 (36.5%) from grade 6. Analyses were then conducted for all valid questionnaires.

### 2.2. System of Scoring

The Education Well-Being Scale developed in this study contained eight concepts, which consisted of four items each, and has a total of 32 questions. Each item was scored using a four-point Likert scale (1 for never, 2 for little, 3 for often, and 4 for always). The higher the total score, the better the perceived education well-being is.

## 3. Results

### 3.1. Reliability Pre-Test Analysis of Education Well-Being Scale

The Cronbach’s α was employed to analyze the reliability of the pre-testing. The scores obtained were: 0.73 for basic need (e.g., Every day is enjoyable); 0.82 for family companionship (e.g., I feel that my family care for me); 0.95 for happy learning (e.g., I feel happy going to school every day); 0.86 for teacher-student interaction (e.g., I feel that the teacher cares for me); 0.77 for safety protection (e.g., The school’s sports facilities are safe to use); 0.84 for school rules (e.g., The school provides rest for students according to the school schedule); 0.98 for peer learning (e.g., I use my recess time with my classmates to complete our own tasks); and 0.77 for peer interaction (e.g., My classmates and I remind each other of our homework and activities to do every day). All scores were greater than 0.7 which indicates that the scale is reliable.

### 3.2. Reliability Formal Analysis of Education Well-Being Scale

Table 1 shows the results of the analyses for the formal test. For parents’ support, the results were: mean = 3.18 to 3.64; standard deviation = 0.60 to 0.74; correlation coefficient = 0.43 to 0.61; and Cronbach’s α = 0.73. For family companionship, the values obtained were: mean = 2.47 to 3.5; standard deviation = 0.65 to 0.98; correlation coefficient = 0.42 to 0.54; and Cronbach’s α = 0.70. For happy learning, the results were: mean = 2.95 to 3.36; standard deviation = 0.74 to 0.87; correlation coefficient = 0.59 to 0.69; and Cronbach’s α = 0.82. For teacher-student interaction the following were obtained: mean = 3.09 to 3.42; standard deviation = 0.73 to 0.94; the correlation coefficient = 0.69 to 0.80; and Cronbach’s α = 0.89. For safety protection the results were: mean = 2.65 to 3.30; standard deviation = 0.77 to 0.91; correlation coefficient = 0.46 to 0.65; and Cronbach’s α = 0.78. For school rules, the values obtained were: 3.36 to 3.59; standard deviation = 0.61 to 0.74; correlation coefficient = 0.59 to 0.70; and Cronbach’s α = 0.82. For peer learning, the results obtained were: mean = 3.24 to 3.61; standard deviation = 0.65 to 0.83; correlation coefficient = 0.48 to 0.67; and Cronbach’s α = 0.77. For peer interaction, the results were: mean = 3.03 to 3.14; standard deviation = 0.86 to 0.92; correlation coefficient = 0.58 to 0.62; and Cronbach’s α = 0.80. The absolute value of skewness was not more than 1, which indicates normal distribution. The reliability of the whole scale was 0.93, which suggests high consistency.

### 3.3. The Validity Analysis for Educational Well-Being Scale

The educational well-being scale was derived from theories based on previous literature and related studies. According to Yu [23], confirmatory factor analysis needs to be directly adopted when testing the structure of a scale to evaluate the degree of compatibility between the data and the theoretical model.

Since the educational well-being scale consisted of 32 items, to reduce the operational complexity and to focus on the discussion of higher-order factors, under the premise of item analysis and good reliability (Cronbach’s α values were between 0.70 and 0.82), this research firstly integrated all items according to their factors. Its average was used as an explicit variable in confirmatory factor analysis [24]. In terms of model setting, because all the concepts were related theoretically, the first-order four-factor orthogonal model, which assumed that all factors were unrelated, was not analyzed and was simply excluded from the theoretical framework. The first detection model assumed that parents’ support, family companionship, school care, and peer assistance were significantly and positively correlated. The second-order single-factor model of the second test assumed that the four potential variables could be further integrated into a complete concept of educational well-being. In terms of the steps taken, the fitness indexes of the two models met the above-mentioned criteria and had acceptable reliability and validity. Therefore, the comparison of the fitness indexes was used to test which model was more supported by the data. According to the suggestions of Tucker and Lewis [25], Byrne [26], and Hu and Bentler [27], the six indicators, including GFI, CFI, TLI, NFI (need > 0.90), RMSEA (need > 0.08), and SRMR (need > 0.05), were used to evaluate the degree of compatibility between the obtained data and the theoretical model. 

In this study, the mean of the eight explicit variables ranged from 3.03 to 3.48, the standard deviation ranged from 0.50 to 0.70, and the zero-order correlation coefficients ranged from 0.38 to 0.66. All of which were positive and significant. The detailed data are shown in Table 2.

Before carrying out the structural equation model, it is necessary to test its normality, which can be divided into univariate and multivariate parts. In the univariate normalcy test, most studies take a specific absolute score as the criterion to judge the skewness coefficient and kurtosis coefficient. In terms of the application of the structural equation model. Kline [28] believes that when the absolute value of the skewness coefficient is greater than 3, and the absolute value of the kurtosis coefficient is greater than 10, the norm is violated. In this study, the absolute value of skewness coefficient and kurtosis coefficient of all the explicit variables were all less than 1, which met the standard. Therefore, the data could be regarded as conformant to the univariate normal distribution. For the multivariate normality, estimate the multivariate kurtosis (also known as the Mardia coefficient) as the test of multivariate normality. The results showed that the Mardia coefficient was less than 10, indicating that the data conformed to the multivariate normal assumption required by the most probable likelihood estimation method. The analysis used IBM SPSS version 21 (IBM SPSS Statistics for Windows, Version 21.0, Asia Analytics Taiwan Ltd., Taipei, Taiwan), and AMOS 20 software (IBM SPSS Amos 20, Asia Analytics Taiwan Ltd., Taipei, Taiwan) was used to conduct the multivariate normal test and subsequent confirmatory factor analysis.

After analysis, the results showed that the first-order four-factor oblique model was well adapted (see Table 3). The second-order single-factor model ($\chi^{2}$ = 153.01, *p* < 0.0001, df = 17, GFI = 0.953, CFI = 0.947, TLI = 0.913, NFI = 0.941, RMSEA = 0.098, SRMR = 0.043) was higher than the standard of 0.08 in RMSEA index and the performances of other indexes were also good. The first-order four-factor oblique pattern ($\chi^{2}$ = 89.09, *p* < 0.001, df =14, GFI = 0.975, CFI = 0.971, TLI = 0.971, NFI = 0.966, RMSEA = 0.078, SRMR = 0.032) met the criteria in all indexes and the correlation among potential variables was between 0.72 to 0.97, which were all significantly positive. The detailed data are shown in Table 4. In comparison, the fitness index showed that the first-order four-factor oblique model was superior to the second-order single-factor model. Therefore, the educational well-being induction could be regarded as a multi-dimensional concept and the scores of each subscale can also be used in the scale scoring.

The results supported that the first-order four-factor oblique model was a better theoretical model, so the following confirmatory factor analysis data were dominated by this model. The average variance extracted (AVE) of each factor was between 0.46 to 0.63 and the component reliability (CR) was between 0.63 to 0.77. All of them met the criteria suggested by Fornell and Larcker [29] and Hair, et al. [30]. The results of the measurement model showed that the educational well-being scale has good reliability and validity. The detailed data are shown in Table 5.

After verifying that the structure of the first-order four-factor oblique pattern was supported by the data, cross-validation analysis was performed to ensure the stability of the data structure (see Table 6). The samples were randomly divided into correction samples and validity samples, and then the nested model was analyzed by the multi-group method. The △$\chi^{2}$ (not significant), △CFI (not more than 0.03), and △RMSEA (not more than 0.015) were used for the analysis [31,32]. The results showed that all of them adhered to the standard, indicating that the rechecking effect of the scale was good.

### 3.4. The Differences in the Educational Well-Being among Students of Grades 5 and 6

According to Table 7, the average scores of grade 5 students on educational well-being *(t* = 3.48, *p* < 0.001), parents’ support (*t* = 2.89, *p* < 0.05), basic need (*t* = 2.36, *p* < 0.05), family companionship (*t* = 2.62, *p* < 0.05), teachers’ care (*t* = 4.62, *p* < 0.001), happy learning (*t* = 4.62, *p* < 0.001), teacher-student interaction (*t* = 3.72, *p* < 0.001), safety protection (t = 3.41, *p* < 0.001), and school rules (*t* =3.03, *p* < 0.05) were all significantly higher than that of grade 6 students. There was little difference on the other factors.

## 4. Discussion

This study developed a tool to evaluate the status of children’s educational well-being by focusing on two aspects: (1) to determine if the scale is a potential variable structure of the second-order single-factor; and (2) to establish the psychometric characteristics of the educational well-being scale. In terms of the structure of potential variables, this study first conducted a confirmatory factor analysis on the measurement model of educational well-being. Overall, the results of the fitness tests were all within a reasonable range, which confirmed that the educational well-being scale was indeed a second-order single-factor potential variable structure. The four sub-factors namely parents’ support, teachers’ care, students’ care in school, and peer cooperation, can be explained separately. Except for the poor explanatory power of some questions, the other measures all had a good weight of construct interpretation (*p* > 0.5); that is, this tool is in line with the intended psychometric characteristics.

The results of grade 5 students on educational well-being, parents’ support, basic need, teachers’ care, happy learning, teacher-student interaction, safety protection, and school rules were all significantly higher than that of grade 6 students.

As one gets older, one’s life experiences increase and accumulate, creating a change in the perception of well-being [21,22]. Some students in grades 5 and 6 were about to enter the adolescence preparation stage and some of them have entered the stage of puberty. Further, the sixth grade is the predecessor of middle school where they transition from the elementary stage to the adolescent stage. The physiological changes, work pressure, and the external environment that these students face vary from each other. Therefore, under these circumstances, it is necessary to understand whether there is a difference in the perception of educational well-being among students of different grade levels.

The biggest difference between the scale developed in this study and the previous domestic well-being scales is that the previous ones were measured from the emotional, psychological, and social levels (e.g., [33]). The items included in this study were mainly derived from the satisfaction of children’s subjective feelings about the nature of educational well-being; that is, children’s feelings about the development and implementation of education. This allowed us to explore and verify the agenda-oriented aspect of well-being and make up for the long-neglected topic of well-being in the past domestic research.

In terms of its practical application, since the educational well-being scale developed in this study is composed of four aspects namely, parents’ support, teachers’ care, students’ care in school, and peer cooperation, it may be used to supplement the index of subjective well-being for future research that intends to employ a cross-sectional or longitudinal survey to understand the perception of individuals receiving formal education on educational well-being. Further, this scale investigates students’ educational well-being in relation to peers, teachers, parents, and schools. Therefore, the different needs of students can be taken into account, diverse perceptions can be considered, and students’ emotional acceptance and response to education can be given attention [34]. Because of the humanistic approach of this scale, it could be used as a reference to improve the educational system to help individuals enhance their well-being experience and ensure its attainment [35]. Lastly, the scale can be used as an evaluation index wherein schools could set a target score that they would need to achieve at the end of every semester to make sure that students have better educational well-being every school year.

Also, the education development scale developed for this research can be used as the course assessment tool for health behavior in school-aged children. From the research conducted by Alex Bertrams from the Department of Educational Psychology at University of Bern showed that well-being courses have positive influence on the subjective well-being of the students [36]. Thus, students’ physical and psychological health and sense of happiness may be enhanced if well-being courses are being promoted in elementary schools.

## 5. Conclusions

### 5.1. The Education Well-Being Scale Has a Stable Factor Structure

This research proved that educational well-being can be explored based on four factors namely parents, teachers, peers, and schools; and from which, indicators can be constructed downwards. This study described parents’ support as to how parents give attention to their children’s basic needs and how family companionship is established at home. Meanwhile, teachers’ level of care was described as how teachers ensure that children are happily learning in school and how teacher-student interactions are made. Students’ care in school includes the students’ adherence to rules and regulations of the school. Lastly, peer cooperation involved how students interact with peers at school. The results of the confirmatory factor analysis (CFA) showed that the educational well-being scale was a potential variable structure for second-order single-factor. The overall fit index illustrated the structure of the four factors included in the 32 items and indicated that the empirical data supports these factors.

### 5.2. The Educational Well-Being Scale Has Good Reliability

The Cronbach’s α coefficients of each factor in the educational well-being scale were between 0.73 to 0.95 and 0.73 to 0.89, and the reliability score of the whole scale was 0.93, which is higher than the score (0.70) suggested by Nunnally and Bernstein [37]. McDonald’s omega as a parameter of internal consistency was 0.93. This means that the scale has good reliability.

### 5.3. Students in Different Grade Levels Had Different Educational Well-Being Scores

The results showed that the grade 5 students had significantly higher scores than grade 6 students in the overall educational well-being score and in the seven aspects (parents’ support, basic need, teachers’ care, happy learning, teacher-student interaction, safety protection, and school rules).

### 5.4. Future Suggestions

This study suggests that future research could collect more valid evidence (e.g., predictive validity, etc.) to further improve the study. Moreover, it is recommended that researchers perform further cross-sample analysis and include junior high school students and high school vocational students to explore educational happiness.

The connotation of “educational well-being” is profound and broad, and it is suggested for future research to engage in more aspects, such as self-esteem, social behavior, self-concept, etc. to further expand the educational well-being scale and help students attain better educational well-being.

## Figures and Tables

**Table 1 healthcare-09-01045-t001:** Statistical analysis for Educational Well-being Scale (*n* = 960).

Item	Title	*M*	*SD*	Skewness	*r*	Cronbach’s α/McDonald’s Omega
Basic need	1. I am in good health.	3.27	0.69	−0.64	0.43	0.73/0.73
	2. Every day is enjoyable.	3.31	0.71	−0.81	0.61	
	3. I feel fresh every day.	3.18	0.74	−0.50	0.60	
	4. I am satisfied with the food, clothing, housing, and transportation in my current life.	3.64	0.60	−1.72	0.46	
Family companionship	5. I feel that my family care for me.	3.59	0.65	−1.55	0.42	0.70/0.70
	6. I ask my family for help when I need it.	3.21	0.83	−0.71	0.54	
	7. My family would accompany me to study or do my homework.	2.47	0.98	0.16	0.54	
	8. My family participates in my school activities.	2.82	0.95	−0.20	0.46	
Happy learning	9. I feel happy going to school every day.	3.16	0.79	−0.70	0.62	0.82/0.82
	10. I expect a lot from school.	2.98	0.87	−0.55	0.69	
	11. The teacher’s class encourages me to participate in learning activities.	2.95	0.85	−0.49	0.68	
	12. Learning at school makes me gain a lot of knowledge.	3.36	0.74	−1.03	0.59	
Teacher-student (T-s) interaction	13. I feel that the teacher cares for me.	3.25	0.84	−0.92	0.80	0.89/0.89
	14. The teacher cares about me as a friend.	3.09	0.94	−0.70	0.80	
	15. The teacher is full of enthusiasm and energy in class.	3.14	0.88	−0.73	0.73	
	16. The teacher puts his or her heart into teaching when he or she is in class.	3.42	0.73	−1.15	0.69	
Safety protection	17. The school nurse (teacher) takes good care of me.	3.30	0.81	−0.91	0.46	0.78/0.78
	18. The school’s sports facilities are safe to use.	3.16	0.77	−0.72	0.62	
	19. The school toilet is well designed and clean, so I can use it with peace of mind.	2.65	0.91	−0.19	0.62	
	20. The campus environment is very safe.	3.16	0.80	−0.71	0.65	
School rules	21. The school provides rest for students according to the school schedule.	3.36	0.74	−0.92	0.59	0.82/0.82
	22. I respect the teachers and follow their instructions.	3.48	0.65	−1.03	0.70	
	23. I comply with the class life convention.	3.41	0.71	−1.03	0.70	
	24. I use the school’s equipment and facilities with utmost care.	3.59	0.61	−1.37	0.61	
Peer learning	25. I get along well with my classmates.	3.49	0.66	−1.12	0.56	0.77/0.77
	26. I discuss my homework with my classmates.	3.24	0.83	−0.74	0.48	
	27. My classmates and I help each other when we need it.	3.52	0.65	−1.18	0.67	
	28. It’s nice to be with my classmates.	3.61	0.65	−1.62	0.59	
Peer interaction	29. My classmates and I remind each other of our homework and activities to do every day.	3.03	0.92	−0.59	0.58	0.80/0.80
	30. I use my recess time with my classmates to complete our own tasks.	3.16	0.86	−0.82	0.62	
	31. Teachers remind students to reduce their time for homework by using auxiliary tools.	3.05	0.90	−0.59	0.61	
	32. I come up with good ways to improve my efficiency when doing my homework with my classmates.	3.14	0.86	−0.69	0.62	

Note: the revised item-total score correlation is used for the correlation coefficient.

**Table 2 healthcare-09-01045-t002:** Explicit variable descriptive statistics and correlation coefficient matrix.

	*M*	*SD*	Skewness	Kurtosis	1	2	3	4	5	6	7	8
1. Basic need	3.35	0.50	−0.62	0.05	−							
2. Family companion-ship	3.03	0.61	−0.34	−0.37	0.50	−						
3. Happy learning	3.14	0.62	−0.51	−0.10	0.58	0.51	−					
4. T-s interaction	3.26	0.70	−0.80	0.09	0.38	0.39	0.66	−				
5. Safety protection	3.09	0.61	−0.49	−0.10	0.48	0.42	0.58	0.53	−			
6. School rules	3.48	0.53	−0.78	−0.11	0.47	0.43	0.57	0.54	0.51	−		
7. Peer learning	3.47	0.53	−0.87	0.12	0.49	0.46	0.48	0.38	0.39	0.51	−	
8. Peer interaction	3.11	0.69	−0.55	−0.36	0.39	0.45	0.48	0.39	0.36	0.52	0.65	−

Note: All correlation coefficients reached a significant level (*p* < 0.001).

**Table 3 healthcare-09-01045-t003:** Summary of the fitness indicator.

Indicator	First-Order Four-Actor Oblique Model	Second-Order Single-Factor Model	Decision Value
*χ* ^2^	89.09 ***	153.01 ***	
df	14.000	17	
GFI	0.975	0.953	>0.90
CFI	0.971	0.947	>0.90
TLI	0.971	0.913	>0.90
NFI	0.966	0.941	>0.90
RMSEA	0.078	0.098	<0.08
SRMR	0.032	0.043	<0.05

Note: *** *p* < 0.001.

**Table 4 healthcare-09-01045-t004:** Potential correlation matrix of the first-order four-factor oblique model.

First-Order Potential Variables	1	2	3	4
1. Parents’ support	-			
2. Teachers’ care	0.89	-		
3. Students in school care	0.69	0.79	-	
4. Peer cooperation	0.79	0.73	0.67	-

Note: All correlation coefficients reached a significant level (*p* < 0.001).

**Table 5 healthcare-09-01045-t005:** Summary of the first-order four-factor oblique model confirmatory factor analysis.

Factor	Explicit Indicators	Standard Factor Load	Standard Error	*t*	AVE	CR
Parents’ support	Basic need	0.68			0.46	0.63
	Family companionship	0.67	0.09	11.45		
Teachers’ care	Happy learning	0.89			0.63	0.77
	T-s interaction	0.69	0.07	14.76		
Students’ care in school	Safety protection	0.70			0.53	0.70
	School rules	0.76	0.08	11.63		
Peer cooperation	Peer learning	0.82			0.63	0.77
	Peer interaction	0.77	0.10	12.56		

**Table 6 healthcare-09-01045-t006:** Summary of the analysis for the effectiveness review.

Model	*χ* ^2^	df	△*χ*^2^	△df	△RMSEA	△CFI
Individual Evaluation						
Review sample (*n* = 480)	54.27 ***	14				
Effectiveness sample (*n* = 480)	44.81 ***	14				
Identity comparison	Review sample: Effectiveness sample					
Referencing pattern	99.08	28				
Factor load identity	106.95	32	1.75 ns.	4	0.002	0.002
Intercept identity	143.23	40	2.92 ns.	8	0.003	0.011
Residual identity	208.60	58	12.12 ns.	18	0.001	0.011

Note: ns. *p* > 0.5; *** *p*< 0.001.

**Table 7 healthcare-09-01045-t007:** Evaluation results on the variable differences between overseas Chinese and Taiwanese Students.

Variables	Categories	*n*	*M*	*SD*	*t*	Difference	95% CI	
							LL	UL
EWB	grade 5	530	107.80	12.86	3.48 ***	3.35	1.46	5.23
	grade 6	304	104.45	14.18				
Parents’ support	grade 5	530	26.69	3.64	2.89 *	0.83	0.27	1.39
	grade 6	304	25.86	4.17				
Basic need	grade 5	530	13.40	2.06	2.36 *	0.36	0.06	0.66
	grade 6	304	13.04	2.26				
Parents’ support	grade 5	530	13.28	2.36	2.62 *	0.46	0.11	0.80
	grade 6	304	12.82	2.53				
Teachers’ care	grade 5	530	27.47	4.19	4.62 ***	1.54	0.89	2.20
	grade 6	304	25.93	4.89				
Happy learning	grade 5	530	13.38	2.46	4.62 ***	0.84	0.48	1.20
	grade 6	304	12.54	2.65				
T-s interaction	grade 5	530	14.10	2.21	3.72 ***	0.70	0.33	1.07
	grade 6	304	13.39	2.82				
Students care in school	grade 5	530	26.63	3.68	1.81	0.53	−0.05	1.10
	grade 6	304	26.11	4.24				
Safety protection	grade 5	530	12.83	2.54	3.41 ***	0.63	0.27	0.99
	grade 6	304	12.20	2.61				
School rules	grade 5	530	14.37	1.83	3.03 *	0.47	0.16	0.77
	grade 6	304	13.90	2.30				
Peer cooperation	grade 5	530	27.00	4.29	1.44	0.45	−0.16	1.06
	grade 6	304	26.55	4.40				
Peer learning	grade 5	530	14.07	2.07	1.53	0.23	−0.07	0.53
	grade 6	304	13.84	2.18				
Peer interaction	grade 5	530	12.93	2.66	1.13	0.22	−0.16	0.60
	grade 6	304	12.71	2.70				

Note: * *p* < 0.05; *** *p* < 0.001.

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
