# Peer review of "The Development and Psychometric Properties of an Education Well-Being Scale among Elementary School Students"

_healthcare, 2021, doi:10.3390/healthcare9081045_

Round 1

Reviewer 1 Report

The manuscript analyzes the development and validation of a scale to measure educational well-being in Taiwan Elementary School students. Overall, it is a rigorous study methodologically and conceptually. Therefore, it can be an interesting contribution, although some modifications and clarifications are necessary:

  • Abstract: Some more keywords must be included.
  • Introduction:
    1. A brief review should be made of the existing instruments for measuring well-being in education (or some of its constituents). Likewise, the contribution of the instrument that is evaluated in this manuscript should be explicitly indicated.
    2. Page 3: The title of section 1.2. seems incorrect, since it does not correspond to the variables operationalized in sections 1.2.1 and 1.2.2.
    3. In some of the operationalized variables (e.g., teacher-student interaction, safety protection, school rules, peer interaction) it is necessary to justify (on the basis of previous research) their relationship with well-being.
    4. The safety protection construct is unclear. It must be defined.
    5. Page 4, line 158: The year of publication of the Hanafin and Brooks reference must be indicated.
  • Tools: An example item must be provided for each variable. However, the ideal would be to include the full scale as an appendix.
  • Results: It is advisable to estimate the McDonald's omega as a parameter of internal consistency.
  • Discussion: It is necessary to point out the health implications of the results, as well as the limitations of the study.

Author Response

Response to Reviewer 1 Comments

Point 1: Abstract: Some more keywords must be included. 

Response 1:

Keywords: education well-being; education well-being scale, psychometric; elementary students

Point 2: Introduction:

        A brief review should be made of the existing instruments for measuring well-being in education (or some of its constituents). Likewise, the contribution of the instrument that is evaluated in this manuscript should be explicitly indicated.

        Page 3: The title of section 1.2. seems incorrect, since it does not correspond to the variables operationalized in sections 1.2.1 and 1.2.2.

        In some of the operationalized variables (e.g., teacher-student interaction, safety protection, school rules, peer interaction) it is necessary to justify (on the basis of previous research) their relationship with well-being.

        The safety protection construct is unclear. It must be defined.

Response 2:

Antonova et al. (2016) conducted a survey on the educational well-being of students in Italy and Russia to assess students' perception of the school environment (how the school helps in improving skills or abilities), their satisfaction with the school environment (such as the interaction with teachers), their freedom from corporal punishment, and the safety protection they receive in school. Hooker (2017) point out that, the role of the teacher that has significant influence on promoting and developing student well-being in an educational setting that is crucial for them to be productive citizens.

Page 3: change

Page 4 : Already made up

Point 3: Tools: An example item must be provided for each variable. However, the ideal would be to include the full scale as an appendix.

Response 3:

The Cronbach's α was employed to analyze the reliability of the pre-testing. The scores obtained were: 0.73 for basic need (ex: Every day is enjoyable); 0.82 for family companionship (ex: I feel that my family care for me); 0.95 for happy learning (ex: I feel happy going to school every day); 0.86 for teacher-student interaction (ex: I feel that the teacher cares for me); 0.77 for safety protection (ex: The school's sports facilities are safe to use); 0.84 for school rules (ex: The school provides rest for students according to the school schedule); 0.98 for peer learning (ex: I use my recess time with my classmates to complete our own tasks); and 0.77 for peer interaction (ex: My classmates and I remind each other of our homework and activities to do every day). All scores were greater than 0.7 which indicates that the scale is reliable.

Point 4: Results: It is advisable to estimate the McDonald's omega as a parameter of internal consistency.

Response 4:

Please see the Table 1

Point 5: Discussion: It is necessary to point out the health implications of the results, as well as the limitations of the study.

Response 5:

Also, the education development scale developed for this research can be used as the course assessment tool for health behavior in school-aged children. From the re-search conducted by Alex Bertrams from the Department of Educational Psychology at University of Bern showed that well-being courses have positive influence on the sub-jective well-being of the students (Bär, 2011). Thus, students’ physical and psychologi-cal health and sense of happiness may be enhanced if well-being courses are being promoted in elementary schools.

Reviewer 2 Report

first of all, thanks for letting me review the manuscript,

it is a good job, that can be improved. Here are some suggestions that I would like the authors to consider

some references are missing in the introduction,

the methodology section should be improved and better explained

the discussion must be completed, it is scarce. In addition, it should connect with the introduction

in the conclusion numerical data should be eliminated

the bibliography standard should be revised

kind regards

Author Response

Point 1: some references are missing in the introduction,

Response 1:

Antonova et al. (2016) conducted a survey on the educational well-being of students in Italy and Russia to assess students' perception of the school environment (how the school helps in improving skills or abilities), their satisfaction with the school environment (such as the interaction with teachers), their freedom from corporal punishment, and the safety protection they receive in school. Hooker (2017) point out that, the role of the teacher that has significant influence on promoting and developing student well-being in an educational setting that is crucial for them to be productive citizens.

Besides, the Glück macht Schule in Austria aim at self-expression and social responsi-bility fulfillment and focus on happiness of life, psychological health, ability to love your-self, nutrition and body health, and exercise and body (Chibici-Revneanu, 2009). Welling-ton College in the UK also offers courses for achieving psychological health, positive rela-tionship, vision, engagement, sustainable development and meaningful life (Morris, 2013). After evaluation previous literature, it is realized that there is a lack of dimensions and tools for assessing educational well-being. As a result, the following indicators are being pro-posed.

Point 2: the methodology section should be improved and better explained

Response 2: Thanks guidance

Point 3: the discussion must be completed, it is scarce. In addition, it should connect with the introduction

Response 3 : Also, the education development scale developed for this research can be used as the course assessment tool for health behavior in school-aged children. From the research conducted by Alex Bertrams from the Department of Educational Psychology at University of Bern showed that well-being courses have positive influence on the subjective well-being of the students (Bär, 2011). Thus, students’ physical and psychological health and sense of happiness may be enhanced if well-being courses are being promoted in elementary schools.

Point 4: in the conclusion numerical data should be eliminated

Response 4:

5.1The education well-being scale has a stable factor structure

This research proved that educational well-being can be explored based on four factors namely parents, teachers, peers, and schools; and from which, indicators can be constructed downwards. This study described parents' support as to how parents give attention to their children's basic needs and how family companionship is established at home. Meanwhile, teachers' level of care was described as how teachers ensure that children are happily learning in school and how teacher-student interactions are made. Students’ care in school includes the students’ adherence to rules and regulations of the school. Lastly, peer cooperation involved how students interact with peers at school. The results of the confirmatory factor analysis (CFA) showed that the educa-tional well-being scale was a potential variable structure for second-order single-factor. The overall fit index illustrated the structure of the four factors included in the 32 items and indicated that the empirical data supports these factors.

The educational well-being scale has good reliability

The Cronbach’s α coefficients of each factor in the educational well-being scale were between 0.73 to 0.95 and 0.73 to 0.89, and the reliability score of the whole scale was 0.93, which is higher than the score (0.70) suggested by Nunnally and Bernstein (1994). McDonald's omega as a parameter of internal consistency was .93. This means that the scale has good reliability.

Point 5: the bibliography standard should be revised

Response 5 :

Hanafin S., & Brooks A.M. (2005). Report on the development of a national set of child well-being indicators. National Children’s Office/TSO.

Round 2

Reviewer 2 Report

the authors have responded and incorporated my suggestions

the article can be published